# Concurrent and Predictive Validity of a Cycloid Loops Copy Task to Assess Handwriting Disorders in Children

**DOI:** 10.3390/children10020305

**Published:** 2023-02-05

**Authors:** Clémence Lopez, Laurence Vaivre-Douret

**Affiliations:** 1Unit 1018-CESP, PsyDev/NTDA Team, National Institute of Health and Medical Research (INSERM), Faculty of Medicine, University of Paris-Saclay, UVSQ, 91190 Villejuif, France; 2Faculty of Health, Department of Medicine Paris Descartes, Université Paris Cité, 75006 Paris, France; 3University Institute of France (Institut Universitaire de France, IUF), CEDEX 5, 14032 Paris, France; 4Department of Child Psychiatry, AP-HP Centre, Necker-Enfants Malades University Hospital, 75015 Paris, France; 5Department of Endocrinology, IMAGINE Institute, Necker-Enfants Malades University Hospital, 75015 Paris, France; 6Clinical Neurodevelopmental Phenotyping Hôpital Universitaire Necker-Enfants Malades, INSERM UMR 1018-CESP, “Neuro-Développement et Troubles des Apprentissages (NTDA)”, Carré Necker Porte N4, 149, rue de Sèvres, 75015 Paris, France

**Keywords:** pre-scriptural loop task, handwriting disorder, digital pen, spatio-temporal and kinematic measures, gesture patterns, predictive validity, children

## Abstract

Handwriting disorders (HDs) are mainly assessed using script or cursive handwriting tasks. The most common is the scale for children’s handwriting, with a French adaptation (BHK). The present study aims to assess the concurrent validity of a pre-scriptural task (copying a line of cycloid loops) with the BHK for the diagnosis of HDs. Thirty-five primary school children (7 females, 28 males) with HD aged 6–11 years were recruited and compared to 331 typically developing children (TDC). Spatial/temporal/kinematic measures were collected using a digital pen on a paper. Posture and inter-segmental writing arm coordination were video recorded. A logistic regression statistical method, including a receiver-operating characteristic curve, was used to assess the ability of the task to predict HD. Gestural patterns were significantly less mature in HDs than in TDC (*p* < 0.05), and associated with poorer quality, less fluid, and slower drawing (*p* < 0.001). Moreover, good correlations between temporal and kinematic measures and the BHK scale were found. Number of strokes, total drawing time, in-air pauses times, and number of velocity peaks showed very good sensitivity (88%) and specificity (74%) to diagnose HDs. Consequently, the cycloid loops task is an easy, robust, and predictive tool for clinicians to identify HDs before the alphabet is mastered.

## 1. Introduction

Handwriting difficulties affect 10 to 30% of school-age children [1,2] and often co-occur with neurodevelopmental disorders (DCD, ADHD…). However, the development of handwriting skills is paramount to educational attainment [3]. There is currently no international consensus on the definition and characteristics of handwriting disorders (HDs). HDs are assessed using more or less standardised tests, with tasks involving script or cursive handwriting, intended for children learning handwriting skills in primary school. The most widely used assessment is the Rapid Method for Assessment of Children’s Handwriting, which has been adopted in many countries [4], including France (Concise Evaluation Scale for Children’s Handwriting, BHK [5]). It only allows an analysis of handwriting from the copying of a text to assess the quality of the handwriting (legibility, spatial organisation), and/or the speed of transcription. It is, therefore, also dependent on children’s language skills, and it is known that inter-rater agreement with the BHK is low [6].

Furthermore, to our knowledge, there is only one test for the detection of handwriting difficulties that can be used for children before the acquisition of handwriting: the Shore Handwriting Screening for Early Handwriting Development (SHS) [7], usable from the age of three, but information on its psychometric properties is lacking, and no normative sample has been developed [6]. Further to this, different studies [8,9,10,11,12,13,14] have shown that the score obtained on a well-known standardised reproduction test involving geometrical shapes, the Development Test of Visual–Motor Integration (VMI [15]), is a good predictor of handwriting skills. The simplest geometrical shapes (horizontal line, vertical line, circle, cross, square, oblique cross, triangle) correspond to geometrical figures children need to master before learning letters [15]. However, even if it gives insights into motor coordination skills, it does not provide any information on the underlying graphomotor processes that are necessary for the development of handwriting (synergic gestural coordination and temporal and kinematic characteristics). A few studies have examined pre-scriptural skills, but they have only focused on the spatial-temporal and kinematic characteristics of loop drawing—horizontal figures of eight [16,17], horizontal, vertical, right oblique, and left oblique lines [18], or loops [19,20]—on graphic tablets, evidencing poorer quality and less fluid drawing in the HD group. Moreover, the battery for the evaluation of handwriting and orthographic competencies in school-aged children [21] includes a task of copying the letters “le” (in cursive script, which is similar to copying loops), but involves a speed constraint. Our recent study [22,23] on a pre-scriptural task, which involved copying a line of cycloid loops with a free digital pen on a half-sheet of paper placed on a table (naturalistic setting), is the only study that allows performance levels in the genesis of handwriting development to be identified in primary school, using grade-related norms involving spatial-temporal and kinematic dynamic values linked to postural and gestural features (inter-segmental coordination of arm movements with five main patterns). This graphomotor task of moving from left to right with an anti-clockwise loop is allied to handwriting, with loops following the same direction as certain letters (b, e, h, l, a, c, d, g, o, q) and links between letters. Therefore, this task showed that drawing a line of loops requires fine motor control skills without interference from the visual representation of letters, and it is predictive of the BHK scores [23]. Thus, the emergence of this segmental and joint coordination of the writing arm is a prerequisite for the development of cursive writing, and the fluidity in movement (spatial-temporal and kinematic measures) helps to gain speed, which means that the automatization of the graphomotor gesture can evidence proactive motor control [24] possibly from the 3rd grade [23]. In addition, a variation of this loop copying task was used in an eyes-closed condition in a study of 35 children with handwriting disorders [25]. This study highlighted the involvement of both proprioceptive/kinaesthetic feedback disorders and a disruptive effect of the visual control on the quality of the pre-scriptural drawings related to children showing kinesthetic memory and visuospatial disabilities. In the present study, we aimed to assess the concurrent validity (with the BHK test) and predictive validity of this pre-scriptural task for the diagnosis of handwriting disorders as a useful, new, practical tool for clinicians. Indeed, copying a line of cycloid loops could be a quick, ecological setting, and easy assessment across different languages, enabling the detection of children at risk before they start to master the alphabet. The analysis of construct validity focused on the task’s ability to discriminate sensitive parameters (gestural, temporal-spatial, and kinematic) between a group of typically developing children (normative data) and a group with handwriting disorders.

## 2. Materials and Methods

### 2.1. Participants

Thirty-five primary schools (grades 1 to 5) children aged 6 years 2 months to 11 years 11 months (mean 8.40 ± 1.70) with handwriting disorders (HD group) participated in the study. They were compared to developmental grade-related normative data obtained from 331 typically developing right-handed school children (TDC group) from a previous study [23].

The exclusion criteria for the study were: prematurity (birth < 37 weeks of amenorrhea), dyslexia and severe language disorder, ADHD (according to the DSM-5 criteria [26]), sensory, visual, neurological or genetic disorders, autism spectrum disorder, psychopathology, motor disorder caused by injury or accident, skipping or repeating a school grade. The institutional research ethics committee of Paris Descartes University approved the study procedures (CER·2018-72) conducted in accordance with the Declaration of Helsinki. All the parents provided written informed consent, and all children gave oral consent to participate in the study. The collected data were anonymised.

### 2.2. Design and Measures

Children with handwriting disorders were identified by their teachers, and their difficulties were confirmed by the analysis of their writing class notebooks. An experienced psychomotor therapist conducted these analyses. Figure 1 presents the study design.

#### 2.2.1. Concise Evaluation Scale for Children’s Handwriting, the BHK Scale

Children’s handwriting level was assessed using a French standardised assessment of handwriting, the BHK scale [5]. This handwriting scale is considered the gold standard for assessing handwriting in primary school children. The task is to copy a text in cursive handwriting for five minutes. The handwriting is analysed with an impairment score evaluating the legibility, a velocity score, and scores on 13 items measuring different handwriting components (steadiness of handwriting, deviations from the conventional forms and positions of letters and words).

#### 2.2.2. Experimental Handwriting Assessments

A pre-scriptural task (copying a line of cycloid loops) was performed by all the children. In the same setting as in the previous study [23], the postural organisation and the inter-segmental coordination of the handwriting arm were video recorded with two video cameras and followed by 2D reconstruction. Spatial-temporal and kinematic measures were collected using an Anoto digital pen, recording at 1/10th of a mm and 1/100th of a second. The pen allowed a free gesture since it was connected to a handwriting analysis software program only after the task (Elian software-Seldage, Version 4.2, http://www.seldage.com, accessed on 24 September 2022). The task was carried out under the most ecological conditions, on a chair in front of a table, with feet flat on the ground, and forearms flat on the table without raising the shoulders. Half an A4-sized sheet of paper was positioned width-wise, aligned straight in front of the child, who was free to move it. The model for the loops, which was identical for all children, was presented on an iPad tablet placed in front of the child (Figure 2). In addition, when the child draws, the pen is light and free, not directly connected to a computer. The sheet of paper is placed directly on the table, in the same way as in a classroom situation. It is only once the child has completed the task that the pen is connected to the computer to extract data.

#### 2.2.3. Postural and Gestural Parameters

Different parameters were analysed by viewing the video recordings: the proximal (head, trunk axis, shoulder, elbow, and forearm) and the distal segments and joints (wrist and fingers) in the coordinated gestural organisation of the drawing process, the postural and gestural organisation in relation to the material (sheet, drawing line, pen). Five patterns were identified for the organisational maturity of inter-segmental coordination of the writing arm displacement in our previous study [23]^,^ with good inter-rater reliability (Cohen’s kappa = 0.9). For example, pattern 1: hand rotation at the wrist; pattern 2: wrist movement in flexion-extension; pattern 3: lateral movement of the trunk; pattern 4: forearm lateral movement; and pattern 5: forearm rotation at the elbow. In addition, observational clinical variables relating to the semiology of the motor characteristics of the gesture (fluidity, control, pressure, synkinesis) were considered (see Vaivre-Douret et al. [23]).

#### 2.2.4. Spatial-Temporal and Kinematic Measures

An Anoto digital pen linked to Elian Research software was used to record spatial/temporal/kinematic parameters (Figure 3). The unlined paper sheet comprised a single set of dots printed in ‘watermark’ mode (Anoto). The pen was presented to the child vertically on the sheet so as not to influence the child’s choice of hand with which to write.

The measures collected with the electronic pen were the following:Spatial parameters:○Total drawing length: length of all strokes.○Average length per stroke: length measured by stroke, a stroke corresponding to continuous lines, without lifting the pen from the sheet of paper.○Drawing width: difference between the rightmost point of the cycloid loops line and the leftmost point of the line.○Drawing height: difference between the highest point of the cycloid loops line and the lowest point.○Degree of inclination of the line: average inclination of the loop line from the horizontal.○Height of the loops: average height of each loop.○Spacing between loops: average of the spaces between each of the loops in the line.○Number of loops: number of loops drawn in the entire line.Temporal parameters:○Number of strokes: number of continuous lines, without lifting the pen from the sheet of paper.○Total drawing time: total time taken by the child to complete the loop line, including both the times the pen traces on the sheet, and the paused times when the pen does not make any traces.○Effective drawing time: tracing time during which the pen is in motion and in contact with the sheet.○Number of on-paper pauses: pauses when the pen is no longer drawing and during which it is in contact with the sheet.○Number of in-air pauses: pauses when the pen is no longer drawing and when it is lifted from the sheet.Kinematic parameters:○Number of velocity peaks: moments of acceleration of the drawing before deceleration.○Average and maximum velocity: ratio of the average/maximum total plot time to the average/maximum total plot length.

### 2.3. Statistical Analysis

The statistical analyses were carried out on R software (version 3.5.3) [27]. The degree of significance retained for all assignments was set at 0.05. In order to ascertain whether there was an association between postural and gestural qualitative variables and whether the children belonged to the HD or TDC group, the χ2 test was used. The Kruskal–Wallis test was used to check whether there was an association between spatial-temporal/kinematic variables and whether the children belonged to the HD or TDC group. Pearson’s correlation test was used to check the association between BHK scores and spatial-temporal/kinematic variables. The predictive validity study was conducted by evaluating the performances of the BHK test and the pre-scriptural task using a logistic regression statistical method, including a receiver-operating characteristic curve (ROC). The logistic regression optimal model selection was performed using the caret + glmnet method with a leave-one-out cross-validation technique. ROC statistical method was used to select the optimal model using the largest value. Sensitivity and specificity values with 95% CIs were used.

## 3. Results

### 3.1. Characteristics of the Sample

The handwriting disorder sample (HD group) included 35 school children with handwriting disorders, 7 girls (20%) and 28 boys (80%), aged 6 years 2 months to 11 years 11 months (mean 8.40 ± 1.70). Thirteen of them (37%) presented dysgraphia on the BHK scale, and twelve (34%) presented more moderate handwriting disorders (Table 1). In contrast, ten (29%) were not identified by the BHK test as presenting any writing disorder (Table 1).

Among the 35 children, six (17%) presented developmental coordination disorder (DCD), and two (6%) had high intellectual potential (according to the DSM-5 criteria [26]). DCD is not significantly associated with a more important handwriting disorder, and the two children with high intellectual potential have a mild writing disorder (not identified by the BHK test).

The gender distribution was significantly different in the HD group compared to the TDC comparison group (chi-2, *p* = 2.042 × 10^−5^). Indeed, in the HD group, 20% were girls (*n* = 7) and 80% were boys (*n* = 28). In the TDC group, the distribution is 59% girls (*n* = 196) versus 41% boys (*n* = 135). However, these proportions are consistent with the prevalence of learning disabilities described for school-aged children. Moreover, in our sample of children with a handwriting disorder, there is no statistically significant difference in the distribution of children according to gender, neither concerning the postural and gestural variables (MANOVA, *p* = 0.97) nor concerning the spatial-temporal and kinematic variables (MANOVA, *p* = 0.75). We, therefore, did not separate the processing of children’s data based on their gender in our analyses.

In the HD group, 89% were right-handed (*n* = 31) and 11% were left-handed (*n* = 4). In the comparison TDC group, 100% (*n* = 331) were right-handed. Although the distribution of handedness differs significantly between the HD and TDC groups (chi-2, *p* = 9.843 × 10^−8^), there is no statistically significant difference in the distribution of children with writing disorders in our sample according to handedness, either concerning postural and gestural variables (MANOVA, *p* = 0.5975) or concerning spatio-temporal and kinematic variables (MANOVA, *p* = 0.2091). Therefore, we will not separate right-handed and left-handed children in the rest of our analyses.

### 3.2. School Grades of Children with Handwriting Disorders Compared Typically Developing Children

Concerning the age match between the two groups, the results of our analysis are as follows. Since none of the children in our samples repeated or skipped a grade, we compared school grades between the two groups. We compared the distribution of school grades between the TDC group and the HD group. Table 2 shows this distribution (expressed as a percentage per group) and the result of the chi-2 test comparing the two groups.

Thus, there is no statistically significant difference in the distribution of school grades by group (TDC or HD).

### 3.3. Postural and Gestural Features of Children with Handwriting Disorders Compared to Typically-Developing Children

The static postural organisation was not significantly different between the HD and TDC groups (head position, *p* = 0.11; trunk position, *p* = 0.89; shoulder elevation, *p* = 0.11; elbow elevation, *p* = 016; forearm elevation, *p* = 0.89; wrist elevation, *p* = 0.20; wrist position, *p* = 0.73; wrist in relation to the axis of the arm, *p* = 0.93; mode of grip on the pen, *p* = 0.68). Children in the HD group (*n* = 35) presented significantly poorer (*p* < 0.05) synergistic coordination of the hand gesture than the TDC group (compared to the norms previously obtained [22,23]) during the cycloid loop copy test. As a result, in the HD group, more mature movement in the rotation of the forearm at the elbow (Pattern 5) seemed less frequent (13% of children in the HD group versus 32% in the TDC group), in favour of a more immature lateral movement of the forearm and the elbow (Pattern 4) (71% in the HD group versus 49% in the TDC group) (*p* = 0.002). Children in the HD group also showed a significantly poorer stability of the wrist, with greater flexion-extension movements when drawing the loops (52% in the HD group vs. 29% in the TDC group) (*p* = 0.01) and a positioning of the fingers too far up the pen (16% vs. 3%) (*p* = 0.02). This immature gestural organization was associated with greater neuromuscular involvement in the graphic activity: hyper-controlled gesture (*p* = 0.004), slowed gesture (*p* = 0.0001) and a more proximal starting point (*p* = 0.0004). Oro-facial synkinesis characteristics were more markedly observed (but were not significant [*p* = 0.39]) in the HD group (55%) than in the TDC group (43%).

### 3.4. Spatial, Temporal and Kinematic Dynamic Measures among Children with Handwriting Disorders Compared to Typically Developing Children

The loops were drawn significantly closer to one another (*p* < 0.001), with less fluid movements (*p* < 0.001), and less rapidly (*p* < 0.001) in the HD group than in the TDC group (Table 3).

### 3.5. Concurrent Validity between the Criteria on the BHK Standardized Scale and the Spatial/Temporal/Kinematic Measures in the HD Group (n = 35)

Table 4 below presents statistically significant correlations between the spatial/temporal/kinematic variables collected by the pen and:-the writing velocity according to the BHK scale (number of characters copied),-the scores on the 13 BHK criteria.

There was no statistically significant correlation between the total BHK raw score and the spatial/temporal/kinematic variables.

As a result, the more rapid the writing on the BHK scale, the greater the speed in drawing a line of loops (decrease in drawing time, *p* < 0.05; increase in the average drawing speed and maximum drawing speed, *p* < 0.05).

The larger the writing on the BHK scale, the higher the loops and the greater the spacing between the loops in the loop assignment (*p* < 0.05).

The greater the inclination of the margin on the BHK scale, the higher the line of loops (increase in the height of the drawing and the height of the loops, *p* < 0.05).

The more chaotic the writing on the BHK scale, the longer the child took to draw the line of loops (increase in the total duration of the task and the effective drawing time, *p* < 0.01; decrease in the mean drawing velocity, *p* < 0.5). This was associated with a significant increase in the number of pauses and their duration (increase in the number of strokes to draw a line of loops and the times in-air pauses, *p* < 0.001; decrease in the mean length per stroke, *p* < 0.05) and with more cramped loops (decrease in the height of the loops and the spacing between the loops, *p* < 0.05).

The greater the number of corrected letters on the BHK scale (higher scores on criterion 12), the greater the number of loops copied in a line (*p* < 0.01) and the smaller the space between them (*p* < 0.05).

The poorer the graphomotor control (more inaccurate drawings, hesitations and shaking), the longer it took for the line of loops to be drawn (increase in the total duration of the task, *p* < 0.01) with a slowing of the drawing process (increase in total drawing time, *p* < 0.01) and more times in-air pauses (*p* < 0.05).

### 3.6. Predictive Validity: Highlighting the Most Discriminating Variables to Diagnose Handwriting Disorders (Logistic Regression Method)

A logistic regression method, including a receiver operating characteristic statistical method (ROC curves), was used to highlight the most discriminating variables to diagnose handwriting disorders. Children with handwriting disorders diagnosed on the BHK scale (*n* = 25; 12 children from the moderate handwriting disorder group and 13 children from the dysgraphia group) were compared with children detected by the BHK scale as good writers (*n* = 132; 10 children from the handwriting disorder not identified by the BHK test group and 122 children from the TDC reference group) (Figure 4). The ROC statistical method is a performance measure of a binary classifier, which allows to determine if a tool is sensitive to classify binary data. In our case, it makes it possible to determine whether the copy test of a line of loops is sensitive to diagnose the presence or absence of a handwriting disorder. ROC was used to select the optimal model using the largest value. The final values used for the model were: alpha = 0.55 and lambda = 0.02823989.

The ROC statistical method allows to obtain an optimal threshold with a very good sensitivity (Se = 88.0%) and a good specificity (Sp = 74.0%) with an accuracy of 76% [CI 95% 68.94–82.94%; *p =* 1.343 × 10^−6^]) and a balanced accuracy of 81% in discriminating between the presence/absence of handwriting disorders.

The logistic regression model using the caret + glmnet method has retained 7 discriminating variables of handwriting disorders. These results are presented in Table 4.

In the Table 5, the results highlighted three variables with significant weight in the model: the number of velocity peaks, the in-air pauses times, and the number of strokes. Thus, these 3 variables are the most discriminating of handwriting disorders and showed a very good sensitivity (88%) and a good specificity (74%) in discriminating between the presence/absence of handwriting disorders.

## 4. Discussion

To understand the origins of handwriting disorders, several recent studies [19,28] have concluded that there is a need for a refined and objective evaluation of graphic arm gesture organisation among children with a handwriting disorder rather than focusing on an evaluation of the legibility of handwriting. This is particularly important because the level of synergic coordination of hand gestures influences the quality and flow of the gesture [23]. The objective of the present study was to demonstrate the usefulness and validity of a pre-scriptural task consisting of copying a line of cycloid loops to diagnose handwriting disorders among primary school children. This task enables an evaluation of the motor and spatial-temporal prerequisites of handwriting without taking account of the child’s cognitive and linguistic abilities. The present results showed that compared to developmental norms previously obtained [23,29], poorer synergistic coordination of the handwriting arm among children with a handwriting disorder was characterised by the persistence, whatever the age, of a progression along the line consisting of moving the forearm and the elbow (pattern 4) (*p* < 0.01) rather than a more mature rotation movement of the forearm at the elbow (pattern 5). The wrist was also less stable (*p* < 0.05) and the hand gesture was slower and hyper-controlled (*p* < 0.01). Alongside, the drawing was poorer in quality, less fluid, and slower (*p* < 0.001). These results are consistent with the study of Gargot et al. [30], which shows a correlation between poorer writing quality and increased time in the air, decreased speed, and steeper accelerations/decelerations and with the study of Casteran et al. [31], which shows longer and slower handwriting (increases of on-paper time, in-air time, and pausing time) in dysgraphic children. This qualitative and quantitative immaturity of the graphomotor gesture detected in a pre-scriptural task of copying a line of loops among children with a handwriting disorder shows how useful this tool could be to detect potential delays in the acquisition of the motor prerequisites of handwriting.

A concurrent validity of this task was analysed using the BHK [5] test for the diagnosis of handwriting disorders. Here, correlations were observed between the spatial/temporal/kinematic variables collected during the loop drawing and the scores obtained on the BHK scale, a measure that is a widely used measure for the evaluation of handwriting disorders in France. The present results showed that strong and significant correlations were observed between the spatial/temporal/kinematic variables during the copying of loops and the scores obtained on the BHK scale.

The predictive validity of the loop copying task was evaluated by its discriminating character for diagnosing handwriting disorders. The results of the logistic regression model are consistent with Gargot et al. [30] and showed that an increase in the number of acceleration peaks, the in-air pauses times and in the number of strokes were the most discriminating measures for handwriting disorders, which confirms the validity of the task. Indeed, these criteria enabled a correct diagnosis for 88% of the children in the study with a handwriting disorder (including those detected on the BHK scale) with a specificity of 74%. In contrast, the spatial measures (length, height and width of the drawings, degree of inclination of the line, space between the loops) and velocity, although they differed between the TDC and the HD group, did not show sufficient sensitivity. This concurs with the results obtained by Asselborn et al. [32], which showed a sensitivity of 78% for the kinematic variables (velocity, number of velocity peak changes per second, and different speed frequencies) recorded on an iPad during the copying of the first BHK paragraph for the diagnosis of handwriting disorders. This raises the issue of the current means of evaluating handwriting disorders and reinforces the importance of completing tests such as the BHK test with other measures. The BHK only assesses velocity and static spatial characteristics in handwriting (see the 13 BHK criteria [4,5]) in terms of performance without reflecting the underlying developmental processes involved in the temporal and kinetic organisation of the drawing, underpinning the ability to use cursive handwriting. This data, however, provides precious information on the rhythmic processes and motor planning of the hand gesture involved in handwriting. One and the same global score on the BHK scale could indeed hide different reasons for handwriting disorders, such as motor or neuro-visual disorders [8,10,33,34,35,36,37] and not necessarily point to dysgraphia, for which treatment will be different.

The good concurrent and predictive validity of this pre-scriptural task involving the copying of a line of cycloid loops to study the graphomotor process involved in handwriting disorders is also confirmed by good inter-rater validity (Cohen’s kappa = 0.9) (refer to Vaivre-Douret et al. [23]). Furthermore, this task, which is simple and quick to administer with a digital pen to capture the dynamics of handwriting on a specific paper sheet, has the advantage of providing developmental gestural landmarks that teachers and clinicians can observe. This task can be administered to children with a cognitive learning disorder, such as dyslexia, or even psychopathological disorders (such as autism spectrum disorders). This task, associated with an analysis of the synergistic gestural organisation of the child, provides important information on children’s graphomotor development and on any difficulties in motor control.

The specific setting of the current study also entails some limitations. First, there was a loss of statistical power because of the small number of HDs. In a future study, it would be interesting to use a multi-parametric statistical approach. Second, the analysis of the video-recorded gestures could have been automated, but the patterns used were nevertheless easily observable. In addition, collecting data from younger children (aged 5 years old to have sufficient access to pre-scripturals) could be interesting to obtain early developmental norms necessary for preventive screening. Finally, it would be interesting to develop the gestural and spatial-temporal/kinematic variables studied in our research also during a handwriting test (in addition to the pre-scriptural loop copy test). For this, it would be useful to examine the common parameters for writing words in different languages to develop a test that could be used internationally.

## 5. Conclusions

To conclude, the graphomotor pre-scriptural task proposed in the present article provides a new, robust tool that shows very good sensitivity (88%) and good specificity (74%) with balanced accuracy of 81% to determine whether or not children are able to master the graphomotor prerequisites giving access to handwriting. The strength of this task is a practical and ecological setting with an automatic capture of dynamic measures using a free digital pen on a specific sheet of paper placed on a table. This differs from studies on graphic tablets, which impose a restricted and rigid setting (no freedom of movement on the table), and a graphic movement that can be hampered by the thickness of the tablet or the way in which the pen comes into contact with the tablet, as demonstrated by several studies [38,39].

This pre-scriptural assessment contributes to solving the problem of the sensitivity of the BHK test, which does not take account of mild handwriting disorders detected by teachers and on exercise books. Thus, our pre-scriptural task enables the detection of children at risk before they start to master the alphabet, and it is useful in clinical decision-making processes for handwriting diagnosis and remediation. Integrated into a global multidimensional evaluation of handwriting disorders, it can contribute to a better understanding of the semiology of the disorders by making it possible to identify delays in the development of the graphomotor gesture and even detect children at risk before they start to master the alphabet.

## Figures and Tables

**Figure 1 children-10-00305-f001:**
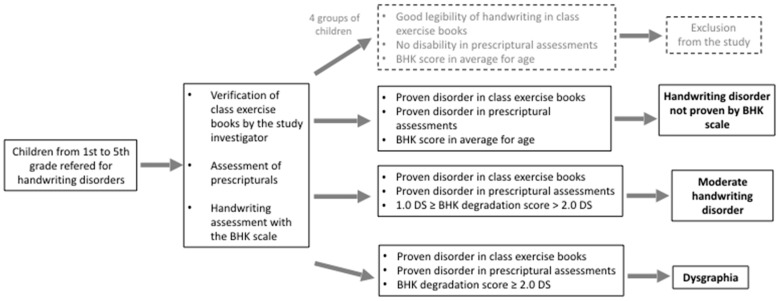
Study design.

**Figure 2 children-10-00305-f002:**
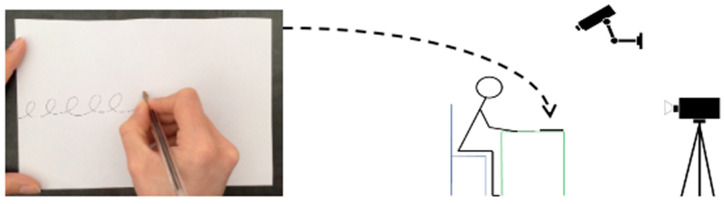
Experimental setting. A video model of a line of cycloid loops was displayed on an iPad placed in front of the child. The experimental features were collected with a video recording system and a digital pen. The task was performed on a free sheet of paper placed directly on the table.

**Figure 3 children-10-00305-f003:**
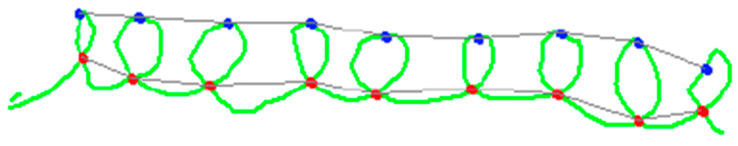
Extracted from Elian Research software (Version 4.2, http://www.seldage.com, accessed on 24 September 2022). Each point forming the line of loops, the highest point of the loop, and the intersection point are recorded for the calculation of the following features: total drawing length, average length per stroke, drawing width, drawing height, degree of inclination of the line, height of the loops, spacing between loops, number of loops, number of strokes, total drawing time, effective drawing time, number of on-paper pauses, number of in-air pauses, number of velocity peaks, average and maximum velocity.

**Figure 4 children-10-00305-f004:**
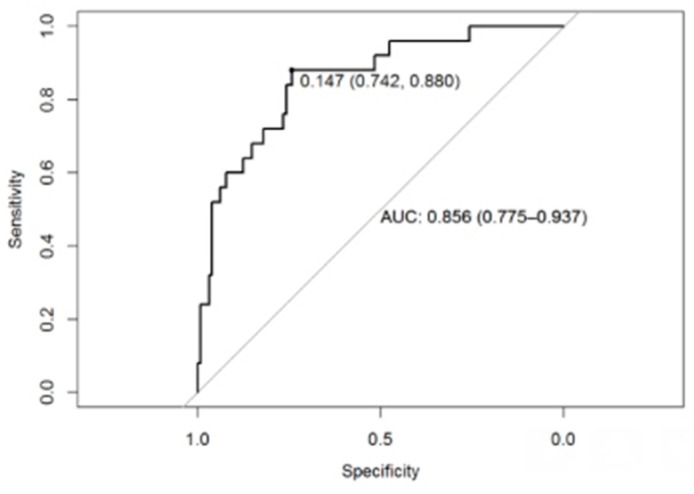
Results of the receiver operating characteristic (ROC curve).

**Table 1 children-10-00305-t001:** Characteristic of the children.

	Handwriting Disorder Not Identified by the BHK Test (*n* = 10)	Moderate Handwriting Disorder (*n* = 12)	Dysgraphia (*n* = 13)	Total (*n* = 35)
BHK degradation score [m (SD)]	−0.12 (0.59)	1.24 (0.33)	2.61 (0.72)	1.41 (1.22)
BHK velocity score [m (SD)]	0.05 (0.54)	0.23 (1.29)	0.11 (1.06)	0.07 (0.998)
Age (months) [m (SD)]	98.7 (17.17)	93.58 (19.06)	108 (21.61)	100.4 (19.99)
Gender [n (F/M)]	1/9	2/10	4/9	7/28

n: number; m: mean; SD: standard deviation; F: female; M: male; BHK degradation score: the lower the degradation score, the better the writing quality.

**Table 2 children-10-00305-t002:** School grades in HD group vs. TDC group.

School Grades	TDC Group (*n* = 331)		HD Group (*n* = 35)		*p*-Value
	*n*	%	*n*	%	
1st grade	58	18	9	26	0.3248
2nd	75	23	8	23	
3rd	72	22	4	11	
4th	64	19	4	11	
5th	62	19	10	29	

*n*: number; %: percentage; TDC: typically developing children; HD: handwriting disorders group.

**Table 3 children-10-00305-t003:** Mean (SD) of spatial, temporal, and kinematic measures for the handwriting disorders group compared to a typical sample instructed to copy a line of cycloid loops.

	Features	Typical Group (*n* = 331)	Handwriting Disorders Group (*n* = 35)	*p*-Value
		m (SD)	m (SD)	
Spatial measures	Total drawing length (mm)	687.5 (166.4)	768.6 (151.7)	0.0018 **
	Average length per stroke (mm)	514.3 (228.7)	314.1 (1.0)	6.123 × 10^−6^ ***
	Drawing width (mm)	194.4 (9.0)	192.8 (13.8)	0.927
	Drawing height (mm)	26.9 (16.6)	24.5 (7.1)	0.760
	Degree of inclination of the line	−1.6 (3.0)	−0.42 (3.4)	0.058
	Height of loops (mm)	12.2 (5.5)	10 (4.4)	0.021 *
	Spacing between loops (mm)	13.3 (6.0)	9.09 (4.1)	9.163 × 10^−6^ ***
	Number of loops	17.9 (9.0)	25.8 (11.4)	2.296 × 10^−5^ ***
Temporal and kinematic measures	Number of strokes	2.0 (2.3)	5.7 (6.7)	7.754 × 10^−9^ ***
	Total drawing time (s)	17.4 (14.1)	33.1 (21.4)	6.683 × 10^−8^ ***
	Effective drawing time (s)	16.09 (12.5)	26.7 (14.2)	5.259 × 10^−7^ ***
	On-paper pauses times (s)	0.12 (0.5)	0.43 (2.2)	0.673
	In-air pauses times (s)	0.87 (3.1)	4.51 (8.8)	9.167 × 10^−6^ ***
	Number of velocity peaks	1.29 (2.4)	2.14 (2.5)	0.003 **
	Average velocity (mm/s)	57.12 (28.1)	37.8 (22.7)	7.594 × 10^−6^ ***
	Maximum velocity (mm/s)	59.1 (27.9)	43.14 (22.4)	0.0002 ***

* *p* < 0.05; ** *p* < 0.01; *** *p* < 0.001; mean; SD: standard deviation.

**Table 4 children-10-00305-t004:** Statistically significant correlations between BHK scale scores and spatial/temporal/kinematic measures for copying a line of cycloid loops (*n* = 35).

BHK Test Criteria	Spatial/Temporal/Kinematic Measures	Pearson’s *r*	*p*-Value	Interpretations
Raw score	No significant correlation			
Number of letters written in 5 mn	Total drawing time (s)	−0.44	0.026 *	Strong negative correlation
	Effective drawing time (s)	−0.45	0.023 *	Strong negative correlation
	Average velocity (mm/s)	0.47	0.017 *	Strong positive correlation
	Maximum velocity (mm/s)	0.44	0.028 *	Strong positive correlation
Criterion 1: writing is too large	Height of loops (mm)	0.49	0.013 *	Strong positive correlation
	Spacing between loops (mm)	0.40	0.049 *	Weak positive correlation
Criterion 2: widening of left margin	Drawing height (mm)	0.43	0.033 *	Strong positive correlation
	Height of loops (mm)	0.44	0.029 *	Strong positive correlation
Criterion 5: acute turns in joining letters	Number of strokes	0.50	0.009 **	Strong positive correlation
	Total drawing time (s)	0.59	0.002 **	Strong positive correlation
	Effective drawing time (s)	0.51	0.009 **	Strong positive correlation
	In-air pauses times (s)	0.52	0.008 **	Strong positive correlation
	Average length per stroke (mm)	−0.41	0.041 *	Strong negative correlation
	Average velocity (mm/s)	−0.41	0.041 *	Strong negative correlation
	Height of loops (mm)	−0.41	0.042 *	Strong negative correlation
	Spacing between loops (mm)	−0.40	0.044 *	Strong negative correlation
Criterion 12: correction of letter forms	Number of loops	0.61	0.001 **	Strong positive correlation
	Spacing between loops (mm)	−0.48	0.016 *	Strong negative correlation
Criterion 13: unsteady writing trace	Total drawing time (s)	0.53	0.006 **	Strong positive correlation
	Effective drawing time (s)	0.51	0.009 **	Strong positive correlation
	In-air pauses times (s)	0.48	0.015 *	Strong positive correlation

**p* < 0.05; ** *p* < 0.01; NS: not significant.

**Table 5 children-10-00305-t005:** Variable importance to diagnose handwriting disorders.

Spatial/Temporal/Kinematic Measures	Overall
Number of velocity peaks	100.0000
In-air pauses times (s)	47.7463
Number of strokes	24.7200
Height of loops (mm)	7.0010
Effective drawing time (s)	4.9866
Average length per stroke (mm)	0.4405
Total drawing length (mm)	0.2584

## Data Availability

The data presented in this study are available on request from the corresponding author.

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
