# Peer review of "Concurrent and Predictive Validity of a Cycloid Loops Copy Task to Assess Handwriting Disorders in Children"

_children, 2023, doi:10.3390/children10020305_

Round 1

Reviewer 1 Report

-          I didn’t see any novelty except the use of cameras, but you use BHK method

-          How you identify children with disorder, and dysgraphia

-          You use the cameras, how you identify the thresholds of their parameters, what are their utilities to identify children with disorder

-          I did not find a comparison with the state of the art

- 3 children:  Too few to identify children with disorder

- Most of the references are old !

Reviewer 2 Report

1. (Lines 99-100) read  "They were compared to developmental grade-related normative data obtained from 331 typically-developing right-handed school children (TDC 100 group) from a previous study [23]". It would be informative if information on the conditions-ecological and otherwise- under which this comparison study was conducted are explained for the reader to evaluate whether the comparison is valid.

2. Line 108 read "All the parents and the children provided written informed 108 consent."  Whereas it is understandable that the parents provided written consent, it is inconceivable that children as young as 6 years 2 months would also have provided written consent given that some of them had writing disorders. Could it be that the children gave assent in one way or another instead of written consent? The authors need to be clear on that.

3.  The characteristics of the sample are inadequately described. For example while it is mentioned that all the children in the comparison group were right handed, there is no breakdown of the HD sample according to handedness. Moreover, the age and gender distributions of the comparison sample are not given. Providing these characteristics is helpful. The reader might be interested in establishing whether the differences in the performances are not because of differences in the sample. Put differently, if there is a statistically significant difference in the age between the HD sample and the TDC sample, then age is confounder.

4. Table 2 can be expanded by adding columns corresponding to columns in Table 1. The HD data has been aggregated into one column. This is good. However, disaggregating the data by adding columns for "Hand writing disorder not detected, moderate HD and disgraphia) would strengthen the analysis. For example the reader would be interested in knowing how the scores for those n= 10 who had HD not detected compare against the TDC group on each of the variables under consideration in the table. Intuitively, the n=10 should be closer to the n=331 TDC than to the others in the HD group.

5. ROC approach. There are two tests being compared which are "line of loops vs BHK scale with the BHK scale being the golden standard. Instead of simply comparing the negative HD vs positive HD using the BHK scale as suggested by the statement "Children with handwriting disorders diagnosed on the BHK scale 302 were compared with children for whom the handwriting disorder was not detected by 303 the BHK scale (29% of children in the HD group)" Lines 302-303, the whole set of results from the loop test should be compared with the BHK scale.

6. How does the model perform when applied to the 331 TDC group? If the model is sound, then the loops test should classify most of them as negative for HD. Since the BHK results are known for all the participants (35+331), it is possible to create an aggregate dataset with all these results and test the extent to which the loops test based on the model made up of most discriminating variables accurately predicts the outcome. There are many packages in R that assist with analysis of predictive accuracy of models.  

Reviewer 3 Report

The paper examines the validity of a prescriptural task (drawing circles) for assessing handwriting disorders. The task is compared with standardized, writing-related measure and its ability to predict a diagnosis of handwriting difficulties is also measured. Thirty-five participants, aged 6-11 years and with disorders were examined and the results were compared to a previously analysed sample of typical children. The results showed that the task is well-suited and perhaps more sensitive than available scriptural tasks to assess handwriting disorders, potentially also in pre-schoolers.

The study is generally well-conducted and language is good enough. I have a few points that should be addressed.

Participants

As far as I realized, no information is provided on the match between the two groups (i.e., a statistical test) for relevant variables (e.g., age).

Design, analysis and results

Figure 1 is labelled “study design”, but I think it is more like inclusion/exclusion criteria in sample selection.

The correspondence between the description of statistical analyses and the results is not clear. Specifically, what in the results represents the chi-square test and the Kruskal-wallis? One table mentions pearson’s correlations, but these are not mentioned when describing the analysis.

In tables 2 and 3, multiple parameters are considered, which increases the probability of obtaining false positives. Were there corrections for multiple comparisons?

Abstract, results and discussion/conclusion

It would help to structure the text according to predictive vs. concurrent validity, since this is the title of the paper.

Discussion/conclusion.

The authors acknowledge the need to collect data in preschoolers (5-year-olds), but I think the crucial step to be taken in order to be able to use this task as a true prescriptural test would be a longitudinal design where younger pre-schoolers (3-5 years) do the drawing task and are later reassessed with writing-related measures.

Round 2

Reviewer 1 Report

Add numbering for sections ?